# Novel Copper(II) Complexes with BIAN Ligands: Synthesis, Structure and Catalytic Properties of the Oxidation of Isopropylbenzene

**Iakov S. Fomenko** [1] , **Olga S. Koshcheeva** [1] , **Nina I. Kuznetsova** [2] , **Tatyana V. Larina** [2] , **Marko I. Gongola** [3] , **Medhanie Afewerki** [3] , **Pavel A. Abramov** [1] , **Alexander S. Novikov** [4,5] and **Artem L. Gushchin** [1,*]

1   Nikolaev Institute of Inorganic Chemistry, Siberian Branch of Russian Academy of Sciences, 3 Acad. Lavrentiev Ave., Novosibirsk 630090, Russia; fom1-93@mail.ru (I.S.F.); koshchee@niic.nsc.ru (O.S.K.); abramov@niic.nsc.ru (P.A.A.)
2   Federal Research Center Boreskov Institute of Catalysis, 5 Acad. Lavrentiev Ave., Novosibirsk 630090, Russia; kuznina@catalysis.nsk.su (N.I.K.); larina@catalysis.ru (T.V.L.)
3   Novosibirsk State University, 1 Pirogova Str., Novosibirsk 630090, Russia; m.gongola@g.nsu.ru (M.I.G.); medahaben@gmail.com (M.A.)
4   Saint Petersburg State University, Universitetskaya Nab. 7/9, Saint Petersburg 199034, Russia; a.s.novikov@spbu.ru
5   Peoples' Friendship University of Russia (RUDN University), Miklukho-Maklaya Street 6, Moscow 117198, Russia
*   Correspondence: gushchin@niic.nsc.ru

**Abstract:** Two new isomeric complexes [CuBr$_2$(R-bian)] (R = 4-Me-Ph (**1**), 2-Me-Ph (**2**)) were obtained by reacting copper(II) bromide with 1,2-bis[(2-methylphenyl)imino]acenaphthene ligands and characterized. The crystal structure of **2** was determined by X-ray diffraction analysis. The copper atom has a distorted square-planar environment; the ω angle between the CuN$_2$ and CuBr$_2$ planes is 37.004°. The calculated ω parameters for optimized structures **1** and **2** were 76.002° and 43.949°, indicating significant deviations from the ideal tetrahedral and square-plane geometries, respectively. Molecules **2** form dimers due to non-covalent Cu···Br contacts, which were analyzed by DFT calculations. The complexes were also characterized by cyclic voltammetry and UV-Vis spectroscopy. A quasi-reversible Cu(II)/Cu(I) redox event with E$_{1/2}$ potentials of 0.81 and 0.66 V (vs. SHE) was found for **1** and **2**, respectively. The electronic absorption spectra showed the presence of Cu(I) species as a result of the partial reduction of the complexes in the acetonitrile solution. Both complexes were tested as homogenous catalysts for the oxidation of isopropylbenzene (IPB) in acetonitrile at low temperatures. Differences in the mechanism of the catalytic reaction and the composition of the reaction products depending on the oxidizing ability of the catalyst were revealed.

**Keywords:** copper complexes; BIAN ligands; redox-active ligands; oxidation; isopropylbenzene

## 1. Introduction

Bis(imino)-acenaphthenes (BIANs) belong to the class of α-diimines, which combine 1,4-diazabutadiene and naphthalene fragments [1–3]. Due to this combination, BIANs have strong σ-donor and π-acceptor properties, providing stabilization of both high and low oxidation states of the metal upon coordination. BIANs form complexes with almost all main group elements [4–8] and transition metals [9–20]. The key feature of BIANs is their pronounced redox activity [21–25], and this property is widely exploited by scientists to implement various catalytic transformations [1]. Historically, the first BIAN-based catalysts were Brookhart's catalysts for the polymerization of olefins [9,26]. The various stereoelectronic properties of BIAN ligands, including their oxidation states, allowed for the modulation of catalyst properties, polyethylene branching, and polymer microstructure [27–29].

Much less attention has been paid to the study of other catalytic processes involving metal/BIAN complexes. The most striking examples are reduction processes. These are hydrogenation [12,30–41], reduction of nitroarenes [42–50], and hydroamination [7,51–55]. Examples of oxidative transformations catalyzed by metal/BIAN complexes are even rarer, possibly due to the electron-withdrawing properties of ligands [56–59]. In particular, V(IV) complexes [VO(acac)(R-bian)]Cl efficiently catalyze the epoxidation of terminal and internal olefins with tert-butyl hydroperoxide or hydrogen peroxide [56], while the related complexes [VOCl$_2$(R-bian)] provide easy CH-oxidation of alkanes with hydrogen peroxide [57,60].

It should be noted that the oxidation reactions catalyzed by BIAN complexes are limited to only one metal, vanadium. Similarly to vanadium(IV), copper(II) is capable of activating peroxides in hydrocarbon oxidation reactions. Indeed, copper(II) complexes with various nitrogen-donor ligands are effective catalysts in the processes of activation (oxygenation) of C-H bonds in saturated hydrocarbons and other organic compounds [61–67]. For example, one of the latest works reported on the high catalytic activity of Cu(II) complexes with N-donor dipinodiazafluorenes in the oxidation of alkanes and alcohols by peroxides [68]. At the same time, there are no data in the literature on the reactivity of copper(II) complexes with BIAN-type ligands in radical chain oxidations. It is only known that complexes of the [CuCl$_2$(R-bian)] type (R = tmp, dpp) catalyze the radical polymerization of styrene in the presence of 2,2′-azobisisobutyronitrile as a free radical initiator [69].

In this work, we synthesized two new copper(II) complexes with redox-active BIAN ligands [CuBr$_2$(R-bian)] (R = 4-Me-Ph (**1**), 2-Me-Ph (**2**)) and studied their structures, electrochemical properties, and reactivity in the oxidation of isopropylbenzene with oxygen. To the best of our knowledge, these are the first examples of the oxygenation of the C-H bond catalyzed by copper-BIAN complexes.

## 2. Results and Discussion

**Synthesis.** Complexes [CuBr$_2$(4-Me-Ph-bian)] (**1**) and [CuBr$_2$(2-Me-Ph-bian)] (**2**) were obtained in high yield by similar procedures by reacting copper(II) bromide with the appropriate BIAN ligand in acetonitrile under reflux conditions (Scheme 1). Earlier works reported the preparation of related complexes [CuCl$_2$(R-bian)] (R = tmp, dpp) [69]. Copper complexes with Me-Ph-bians have not been previously described. Complexes **1** and **2** are structural isomers with respect to each other and differ in the position of the methyl group in the aryl ring of the bian. They have different solubilities. Complex **2** is highly soluble in both dichloromethane and acetonitrile, while complex **1** is less soluble in these solvents.

**Scheme 1.** Synthetic routs towards complexes **1** and **2**.

The IR spectra of both complexes are very similar. IR spectra of **1** and **2** show typical vibration bands of the CH group in the region of 3046–2863 cm$^{-1}$ (for **1**) and 3056–2857 cm$^{-1}$ (for **2**), as well as ν(C-C) and ν(C-N) bands in the 1624–1044 cm$^{-1}$ (for **1**) and 1634–1049 cm$^{-1}$ (for **2**) range.

**Crystal structure.** Single crystals of [CuBr$_2$(2-Me-Ph-bian)] (**2**) suitable for X-ray diffraction were obtained by layering hexane on a solution of **2** in dichloromethane (Table S1 in the Supplementary Materials). Unfortunately, our attempts to crystallize complex **1** were unsuccessful. The molecular structure of **2** is shown in Figure 1. The coordination environment around copper(II) center is a distorted square-planar, which is defined by two N and two Br atoms. The ω angle between the CuN$_2$ and CuBr$_2$ planes is 37.004°, where

$\omega = 0°$ for a perfect square-planar coordination, whereas for a tetrahedral one, $\omega = 90°$ [69]. Thus, the geometry of the coordination node for **2** should be considered distorted square-planar. A high degree of distortion is confirmed by DFT calculations (see below). On the contrary, less distorted structures were found for related complexes: [CuCl$_2$(tmp-bian)] ($\omega = 23.67°$) and [CuCl$_2$(dpp-bian)] ($\omega = 1.54°$) [69]. The aryl substituents are not co-directed relative to each other, forming a dihedral angle between the planes of the phenyl rings equal to 30.040(5)°. The bond lengths of the C = N and C-C are 1.2803(2) and 1.5015(2) Å indicating the neutral state of the 2-Me-C$_6$H$_4$-bian. The 2-Me-C$_6$H$_4$-bian ligand is coordinated symmetrically, the Cu-N bond lengths are 2.026(2) and 2.039(3) Å. In the [CuCl$_2$(tmp-bian)] and [CuCl$_2$(dpp-bian)] complexes, these distances are 2.048 and 2.051 Å, respectively [69]. The Cu-Br bond lengths in **2** are slightly different (2.3517(6) and 2.3385(6)Å). Cu-Br distances are in the range of 2.354–2.391 Å in Cu(II) complexes with dipinodiazafluorenes [68].

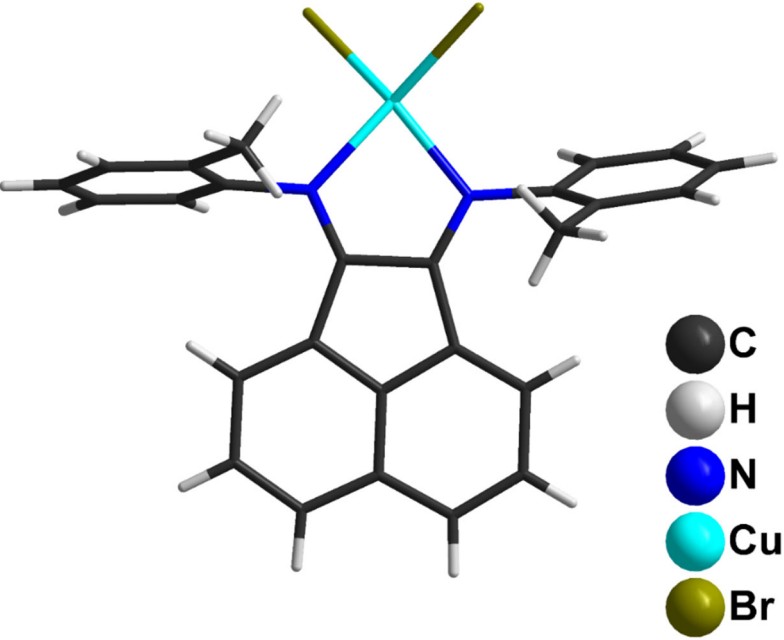

**Figure 1.** Molecular structure of complex **2**. Main bond lengths (Å) and angles (°): C = N, 1.2805(2) and 1.2801(2); C-C, 1.5015(2); Cu-N, 2.026(2) and 2.039(3); Cu-Br, 2.3517(6) and 2.3385(6); N1-Cu-N2, 81.577(7); Br1-Cu-Br2, 99.419(7); N1-Cu-Br2, 95.083(7) and 95.083(7); N1-Cu-Br1, 149.778(8) and 155.648(8).

An interesting feature of crystal packing **2** is the formation of weakly bounded dimers (Figure 2) with intermolecular contacts Cu···Br at 3.666 Å. These interactions were analyzed by DFT calculations (see below).These dimers are connected into infinite supramolecular chains via π–π interactions (Figure S1 in the Supplementary Materials). The chains are directed in the [100] crystal direction (Figure S2 in the Supplementary Materials).

**DFT calculations.** The lack of experimental X-ray structural data for complex **1** prompted us to perform DFT calculations and a full geometry optimization procedure for model structures **1** and **2** in order to compare the geometry of the coordination environment around copper(II) at least in idealized model systems and calculate the corresponding $\omega$ parameters. Results of our DFT calculations reveal that $\omega$ angles between the CuN$_2$ and CuBr$_2$ planes in optimized equilibrium model structures **1** and **2** are 76.002° and 43.949°, respectively. Thus, the optimized equilibrium model structure of **1** could be considered distorted tetrahedral, whereas the optimized equilibrium model structure of **2** is distorted square-planar and close to the experimental X-ray structure ($\omega = 37.004°$). In both cases, the geometry of the coordination CuBr$_2$N$_2$ node is far from ideal (tetrahedral or square-planar), and a strong distortion typical of Cu(II) takes place.

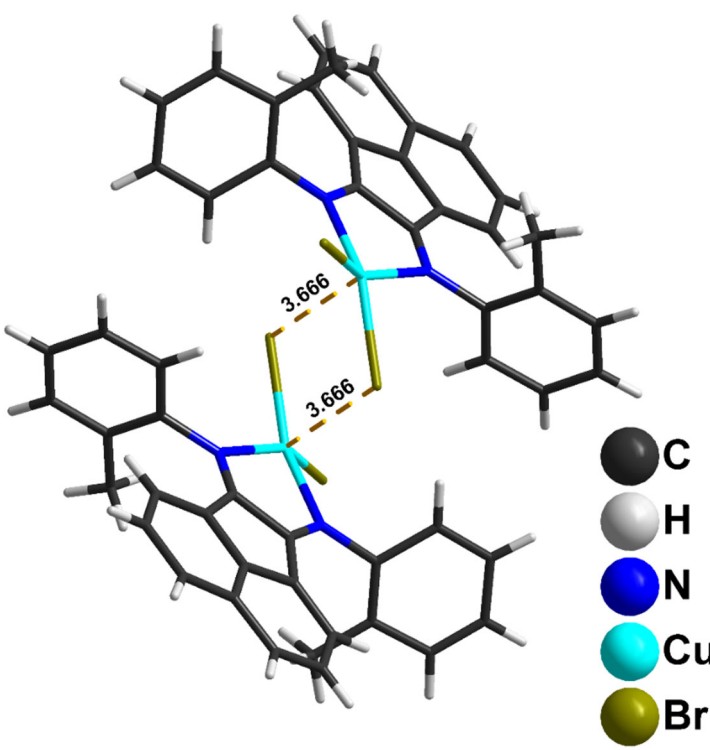

**Figure 2.** Electrostatic dimers in the crystal structure of **2**. Cu···Br contacts are shown with dashed lines.

Inspection of the crystallographic data revealed the presence of interesting intermolecular noncovalent interactions Cu···Br (3.666 Å) in the X-ray structure **2**. In order to confirm or disprove the hypothesis on the existence of these intermolecular noncovalent interactions in the solid state and approximately quantify their strength from a theoretical viewpoint, the DFT calculations followed by the topological analysis of the electron density distribution were carried out at the ωB97XD/DZP-DKH level of theory for model supramolecular associates (see Computational details and Tables S2 and S3 in the Supplementary Materials). Results of topological analysis of the electron density distribution are summarized in Table 1 (the Poincare–Hopf relationship was satisfied), the contour line diagram of the Laplacian of electron density distribution $\nabla^2\rho(\mathbf{r})$, bond paths, and selected zero-flux surfaces, visualization of electron localization function (ELF), and reduced density gradient (RDG) analyses for intermolecular noncovalent interactions Cu···Br in **2** are shown in Figure 3.

**Table 1.** Values of the density of all electrons—$\rho(\mathbf{r})$, Laplacian of electron density—$\nabla^2\rho(\mathbf{r})$ and appropriate $\lambda_2$ eigenvalues, energy density—$H_b$, potential energy density—$V(\mathbf{r})$, Lagrangian kinetic energy—$G(\mathbf{r})$, and electron localization function—ELF (a.u.) at the bond critical point (3, −1), corresponding to intermolecular noncovalent interactions Cu···Br in the X-ray structure **2** and estimated strength for these weak contacts $E_{int}$ [70] (kcal/mol).

| Noncovalent Interaction | $\rho(\mathbf{r})$ | $\nabla^2\rho(\mathbf{r})$ | $\lambda_2$ | $H_b$ | $V(\mathbf{r})$ | $G(\mathbf{r})$ | ELF | $E_{int} \approx -V(\mathbf{r})/2$ |
|---|---|---|---|---|---|---|---|---|
| | | | | **2** | | | | |
| Cu···Br, 3.666 Å | 0.006 | 0.018 | −0.006 | 0.000 | −0.004 | 0.004 | 0.020 | 1.3 |

The topological analysis of the electron density distribution in the model supramolecular associate demonstrates the presence of bond critical point (3, −1) for Cu···Br interactions (Table 1). The low magnitude of the electron density (0.006 a.u.), positive value of the Laplacian of electron density (0.018 a.u.), and zero energy density in this bond critical point

(3, −1) are typical for such weak contacts in similar chemical systems [71–78]. The balance between the Lagrangian kinetic energy G(**r**) and potential energy density V(**r**) at the bond critical point (3, −1) (viz. −G(**r**)/V(**r**) ≥ 1) indicates the absence of covalent contribution in these weak contacts [79]. The Laplacian of electron density is typically decomposed into the sum of contributions along the three principal axes of maximal variation, giving the three eigenvalues of the Hessian matrix ($\lambda_1$, $\lambda_2$ and $\lambda_3$), and the sign of $\lambda_2$ can be utilized to distinguish bonding (attractive, $\lambda_2 < 0$) weak interactions from non-bonding ones (repulsive, $\lambda_2 > 0$) [80,81]. Thus, intermolecular noncovalent Cu···Br interactions are attractive.

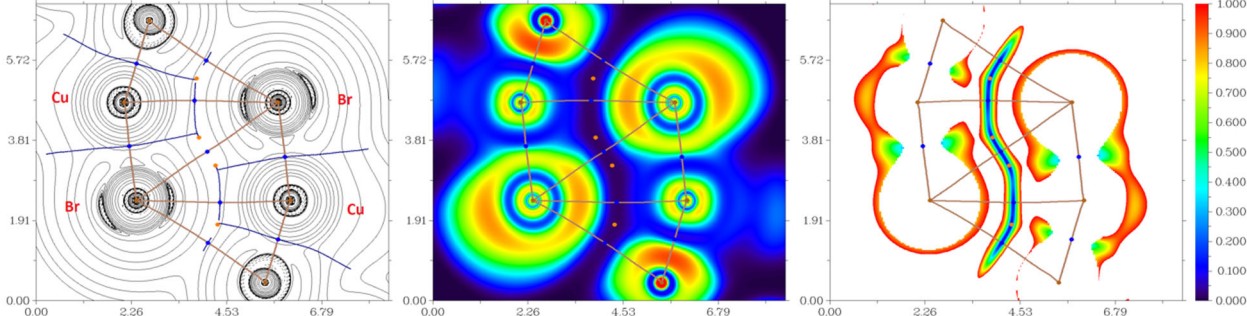

**Figure 3.** Contour line diagram of the Laplacian of electron density distribution $\nabla^2\rho(\mathbf{r})$, bond paths, and selected zero-flux surfaces (left panel), visualization of electron localization function (ELF, center panel), and reduced density gradient (RDG, right panel) analyses for intermolecular noncovalent interactions Cu···Br in the X-ray structure **2**. Bond critical points (3, −1) are shown in blue, nuclear critical points (3, −3)—in pale brown, ring critical points (3, +1)—in orange, bond paths are shown as pale brown lines, length units—Å, and the color scale for the ELF and RDG maps is presented in a.u.

**UV-Vis spectra.** The diffuse reflectance spectra (converted to absorption spectra) of solid samples are characterized by broad bands in the high and low energy regions. (Figure S3). In the high energy region of the spectrum of **1**, there is a band at 315 nm as well as a broad band in the range of 370–550 nm (the edge of this band extends to 680 nm). A poorly resolved wide band in the range of 720–850 nm was found in the low energy region. The spectrum of **2** demonstrates a slightly different pattern. Similar to complex **1**, there is a band at 315 nm and a shoulder at 418 nm. An almost linear increase in absorption with a maximum at 700 nm was detected in the region of 450–700 nm. The high-energy bands in both spectra can be attributed to the π–π* and n–π* transitions of the BIAN ligand. The low energy bands correspond to d-d transitions of the Cu(II) ion ($3d^9$). The hypsochromic shift of the d-d band found for **2** reflects the different coordination environment of the Cu(II) center in the structures (see above).

The UV-Vis spectra of **1** and **2** in acetonitrile solution (Figure S4) show similar absorption bands in the low and high energy regions as for solid samples. Differences appear in the visible region and are associated with the appearance of absorption in the range of 500–700 nm for both complexes. This absorption is especially pronounced for complex **1** with a maximum at 630 nm. We assume that the appearance of a new band is associated with the formation of the Cu(I) species due to the partial reduction of Cu(II). Absorption in this region is most likely associated with MLCT transitions, which are characteristic of Cu(I) complexes [19,82–85]. Apparently, complex **1** is involved in this redox transformation to a greater extent than **2**. This is consistent with electrochemical data, according to which complex **1** is a stronger oxidizing agent than complex **2** (see below).

**Cyclic voltammetry (CV).** In order to determine the redox potential of the Cu(II)/Cu(I) couple, cyclic voltammograms of **1** and **2** in acetonitrile were recorded (Figure S5). A quasi-reversible Cu(II)/Cu(I) redox process centered at $E_{1/2}$ = 0.62 (ΔE = 96 mV) and 0.47 V (ΔE = 137 mV) (vs. Ag/AgCl) or 0.81 and 0.66 V (vs. SHE) was found for **1** and **2**, respectively. An irreversible reduction at 0.49 V and 0.44 V (vs. SHE) was previously reported for the related complexes [CuCl₂(tmp-bian)] and [CuCl₂(dpp-bian)], respectively [69]. The authors attribute the irreversibility of this process to the instability of the reduction product

in terms of Cl⁻ elimination. In our case, the reduction is quasi-reversible, which indicates a higher stability of the reduced Cu(I) species. Note that the Cu(II)/Cu(I) potential for **1** is noticeably shifted to a more positive region compared to **2**, which indicates a higher oxidizing ability of Cu(II) in **1**.

**Catalytic studies.** Both complexes **1** and **2** were tested as homogeneous catalysts for the oxidation of IPB with oxygen at low temperatures (30–50 °C). Oxygen uptake for **1** and **2** began without an induction period (Figure 4), although its presence is typical of many aerobic hydrocarbon oxidations. In solution **1**, a stationary oxygen uptake was established immediately, while in solution **2**, the initial intense uptake gradually slowed down to a stationary value. The steady-state rate of the reaction increased with an increase in the temperature of the experiment (inset in Figure 4). For comparison, oxygen consumption by solutions of 4-Me-Ph-bian (L1) or 2-Me-Ph-bian (L2) developed slowly and differed little from that of the control experiment in the absence of any catalyst (Figure 4, curves L1 and L2). In contrast to complexes **1** and **2**, the CuBr₂-catalyzed oxidation of IPB at 30 °C began after an induction period (Figure S6A) associated with the slow reduction of Cu(II) to Cu(I) [86]. The harder-to-reduce CuCl₂ exhibited a long induction period at 30 °C and higher temperatures (Figure S6B).

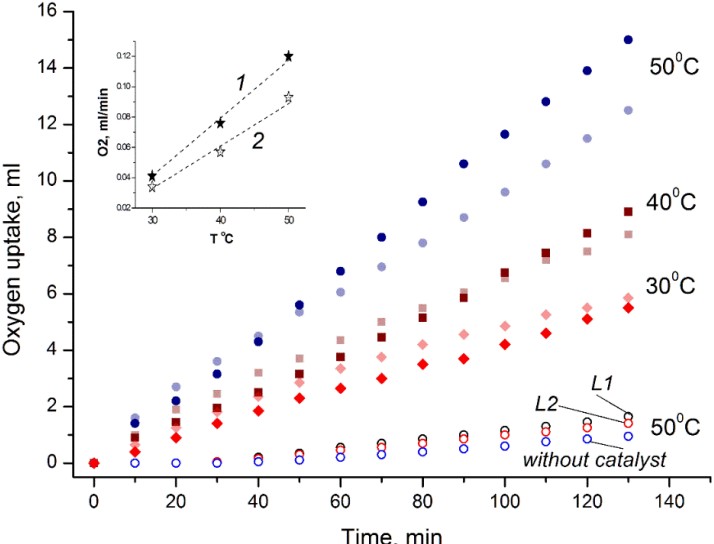

**Figure 4.** Kinetic curves of the oxygen uptake by solutions of **1** (bright symbols), **2** (light symbols), L1, L2 and in the absence of catalyst. Conditions: 1 mL of IPB, 1.5 mL of CH₃CN, $3 \times 10^{-4}$ M catalyst.

The catalysis of the aerobic oxidation of hydrocarbons by transition metal compounds is based on the initiation of radical chains. In our case, the oxidation of IPB in acetonitrile solutions of **1** and **2** is initiated by the decomposition of isopropylbenzene hydroperoxide (IPBHP) due to the Cu(II)/Cu(I) redox couple with the formation of radicals RO• and ROO• (Equations (1) and (2)). The Cu(II)/Cu(I) redox process seems to be facilitated by the presence of the redox-active BIAN ligand, which ensures the reversibility of this process and a rather high oxidizing ability of Cu(II) in **1** and **2** (see CV part). This favors Equations (1) and (2). The initial rate of oxygen consumption by solution of **2** was slightly higher than by **1**. (Figure 4). It decreases with time to a stationary value. IPBHP is contained in a small amount in the initial reaction solution (0.03 mol.%) and accumulates during cycling (Equations (3)–(5)). Termination of the radical chains is usually regarded as bimolecular interactions or self-termination [87]. During the stationary reaction, the rates are close in magnitude for **1** and **2**, with some advantage for **1**, which is especially noticeable at higher temperatures.

$$Cu(I) + ROOH = Cu(II) + RO\bullet + OH\text{-} \qquad (1)$$

$$Cu(II) + ROOH = Cu(I) + ROO\bullet + H+ \tag{2}$$

$$RO\bullet + RH = R\bullet + ROH \tag{3}$$

$$ROO\bullet + RH = R\bullet + ROOH \tag{4}$$

$$R\dot{}\bullet + O_2 = ROO\bullet \tag{5}$$

The UV-Vis spectra show the redox transformations of the complexes during catalysis. The spectrum of the initial solution of **2** in acetonitrile (Figure 5B, black line) does not change after the addition of IPB at room temperature, while significant changes occur during the oxidation of IPB under catalytic conditions (Figure 5B, red line). The absorption in the region of the MLCT transitions of Cu(I) (approx. 530 nm) decreases. On the contrary, the absorption in the region of the d–d transitions of Cu(II) (approx. 680 nm) increases. In contrast to **2**, the spectrum of the initial solution of **1** (Figure 5A, black line) changes after the addition of IPB at room temperature (Figure 5A, green line). The decrease in Cu(II) absorption at approx. 800 nm is accompanied by the appearance of an intense Cu(I) absorption band at 605 nm. Under catalytic conditions, absorption in the region of MLCT transitions of Cu(I) (Figure 5A, red line) is retained.

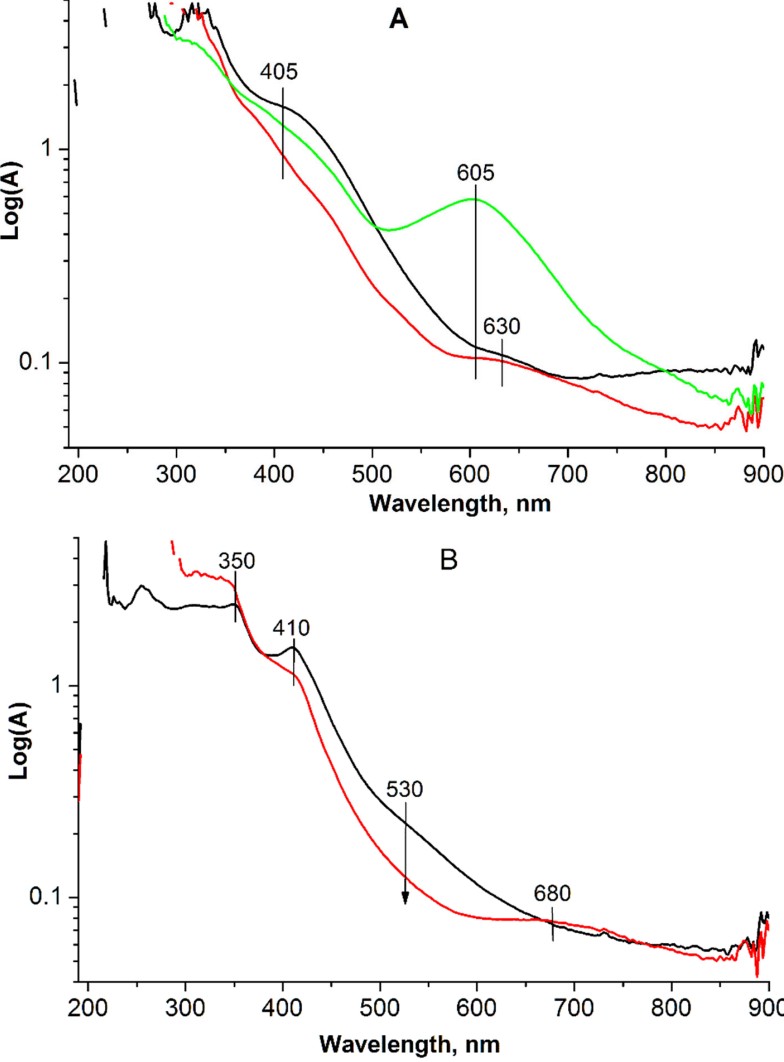

**Figure 5.** UV-Vis spectra of solutions **1** (**A**) and **2** (**B**) in CH$_3$CN ($3 \times 10^{-4}$ M): initial (black spectra), reacted with IPB for 50 min at 40 °C (red spectra) or 10 min at 25 °C (green spectrum).

A similar trend was observed in the spectra of solutions **1** and **2** after the addition of IPBHP (Figure S7). The content of Cu(II) increases in solution **2** (Figure S7B) and decreases in solution **1** (Figure S7A). This is in line with the higher oxidizing ability of complex **1**. Thus, complexes **1** and **2** demonstrate a facile change in the oxidation state at the copper center during reactions both with IPB and IPBHP, wherein both procedures gave spectrally identical results.

According to the radical chain mechanism (Equations (1)–(5)), the oxidation of IPB should give mostly IPBHP and less 1-phenyl-1-propanol (PP). However, GC analysis together with iodometric titration of solutions **1** and **2** after catalysis revealed the formation of IPBHP, PP, as well as acetophenone (AP) and alpha-methylstyrene (AMS) (Scheme 2 and Table 2).

**Scheme 2.** Scheme of IPB oxidation catalyzed by complexes **1**–**2**.

**Table 2.** Results of IPB oxidation in $CH_3CN$ solutions of **1** and **2** (*).

| Complex | T °C | Products, mmol | TON, h$^{-1}$ | Selectivity, % | | |
| --- | --- | --- | --- | --- | --- | --- |
| | | | | IPBHP | PP + AMS (**) | AP |
| | 50 | 0.82 | 504 | 12.2 | 56.3 | 31.5 |
| 1 | 40 | 0.5 | 307 | 16.7 | 58.8 | 24.5 |
| | 30 | 0.31 | 190 | 19.5 | 57.1 | 23.4 |
| | 50 | 0.64 | 393 | 20.3 | 48.2 | 31.5 |
| 2 | 40 | 0.41 | 252 | 22.6 | 53.1 | 24.3 |
| | 30 | 0.32 | 197 | 23.2 | 57.1 | 19.7 |

(*) The conditions are given in the caption to Figure 4. (**) Admixture of AMS appeared from dehydration of PP in the GC column.

The selectivity for IPBHP turned out to be quite low even at 30 °C, and it further decreased as the oxidation temperature increased to 40 and 50 °C (Table 2). While the Cu(II)/Cu(I) redox couples initiate the formation of IPBHP (Equations (1)–(5)), the Cu(II) complexes catalyze the subsequent conversion of IPBHP. It has been confirmed that IPBHP reacts with 1 and 2 in $CH_3CN$ solutions to form PP and AP at 30 °C (Table S4). The same products were previously obtained by decomposition of Cu(II)-IPB peroxo complexes or by decomposition of IPBHP catalyzed by Cu(II) complexes [88–90]. It seems reasonable to suggest the catalyzed by **1** and **2** decomposition of IPBHP follows the same mechanism (Equations (6)–(9)).

$$Cu(II) + ROOH = Cu^{II}\text{-}OH\bullet + RO\bullet \tag{6}$$

$$RO\bullet + RH = ROH + R\bullet \tag{7}$$

$$RO\bullet = AP + CH_3\bullet \tag{8}$$

$$Cu^{II}\text{-}OH\bullet + RH = Cu(II) + H_2O + R\bullet \tag{9}$$

The homolytic O-O bond cleavage of the peroxide moiety (Equation (6)) produces IPB-oxyl radical to form PP (Equation (7)) and AP (Equation (8)) via $CH_3$-scission, in contrast to a less common for Cu(II) complexes heterolytic mechanism, which gives only

PP [91]. Calculations of energetics confirm the homolytic cleavage to be favorable for the decomposition of any Cu(II)-alkylperoxo complexes [92], and comparable to the reductive cleavage of the $Cu^{II}$-O to form Cu(I) (similar to Equation (1)). According to the data of Table 2, the ratio of PP and AP varied within 1.5–2.8, that could be interpreted as indicator of the homolytic mechanism. Homolytic decomposition of IPBHP through the breaking of the O-O bond (Equation (6)) results in formation of $Cu^{II}$-OH• adduct that can participate in the activation of the C-H bond of hydrocarbons [93]. Apart from Equations (1) and (2), the initiation of the cyclic oxidation is intensified by Equations (6)–(9) during the oxidation of IPB. Comparing the catalytic activity of solutions **1** and **2**, we can conclude that a more intense decomposition of IPBHP by complex **1** (Table S4) causes a lower selectivity for IPBHP, but a more intense oxidation of IPB.

## 3. Experimental Section

**General Procedures.** All the experiments were carried out in air. Commercially available reagents ($CuCl_2.2H_2O$, $CuBr_2$, $PPh_3$, chlorobenzene) were used as purchased. 1,2-Bis[(2-methylphenyl)imino]acenaphthene (2-Me-Ph-bian) and 1,2-Bis[(4-methylphenyl) imino]acenaphthene (4-Me-Ph-bian) were prepared as reported [94]. Organic solvents ($CH_2Cl_2$, MeCN, glacial acetic acid, and hexane) were dried by standard methods before use. Cumene (99.9% purity, <0.0003 mmol/mL $\alpha$-methylstyrene) was kindly donated by the Omskiy Caoutchouc factory.

**Physical Measurements**. Elemental C, H, N analyses were performed with a EuroEA3000 Eurovector analyzer. IR spectra were recorded in the 4000–300 $cm^{-1}$ range with a Perkin–Elmer System 2000 FTIR spectrometer with samples in KBr pellets. A Shimadzu UV-2501 PC spectrometer equipped with an ISR-240A diffuse reflectance unit was used for UV-Vis spectroscopy of both solutions and solid samples. Diffuse reflectance (DR) spectra of the solid samples were recorded using $BaSO_4$ as the reflectance standard in the wavelength range of 190–900 nm at room temperature. Samples (fraction < 0.25 mm) were placed in a quartz cuvette with a 2 mm optical path length. The obtained reflectance coefficients were transformed into absorption coefficients using the Kubelka–Munk function, $F(R_\infty) = (1-R_\infty)^2/2R_\infty$. All DR spectroscopy data are presented in the following coordinates: Kubelka–Munk function, $F(R_\infty)$ versus wavelength. The transmission spectra of the solutions were recorded relative to $CH_3CN$ in the wavelength range of 190–900 nm at room temperature. Samples of the solutions were placed in a quartz cuvette with a 1 cm optical path length. The obtained transmission coefficients were transformed into absorption coefficients using the Bouguer–Lambert–Beer law, $A = -\log(I/I_0) = -\log T = \varepsilon l C$. All absorption spectra are presented in the following coordinates: optical density, A or lgA versus wavelength.

**Catalytic Experiments.** The catalytic tests were carried out in a 25 mL glass reactor with a water heating jacket, connected to a thermostat (Huber ministat) by silicone tubes. The reactor had two valves for connection to a graduated gas burette and an oxygen cylinder. The reactor was loaded with $5 \times 10^{-4}$ M acetonitrile solution of a catalyst (1.5 mL) and isopropylbenzene (IPB, 1 mL). The reactor, together with the burette, was purged with $O_2$ at room temperature, sealed, and heated up to a desired temperature under intensive agitation by a magnetic stirrer. When the temperature and pressure stabilized, monitoring of $O_2$ uptake started. After 130 min, the process was terminated, the reactor was quickly cooled to ~15 °C, and the contents were adjusted to 10 mL with acetonitrile and chlorobenzene (internal standard).

The amount of hydroperoxide in the reaction solutions was determined by standard iodometric titration. GC analysis of IPB oxidation products was undertaken on Agilent 7890B instrument equipped with FID and a SOLGEL-WAX 30 m × 0.25 mm × 1 μm column. Prior to GC, the solution was treated with $PPh_3$ to reduce the isopropylbenzenehydroperoxide (IPBHP) to 2-phenyl-2-propanol (PP). Product concentrations were quantified by the internal standard method. The amount of PP formed during the catalytic oxidation was cal-

culated as a difference between the data from GC and iodometric titration. Acetophenone (AP) and α-methylstyrene (AMS) were determined from appropriate GC signals.

**X-ray Crystallography**. Crystallographic data and refinement details for **2** are given in Table S1. The diffraction data were collected on a Bruker D8 Venture diffractometer with a CMOS PHOTON III detector and IμS 3.0 source (Mo Kα radiation, λ = 0.71073 Å) at 150 K. The φ- and ω-scan techniques were employed. Absorption correction was applied by SADABS (Bruker Apex3 software suite: Apex3, SADABS-2016/2 and SAINT, version 2018.7-2; Bruker AXS Inc.: Madison, WI, USA, 2017). Structures were solved by SHELXT [95] and refined by full-matrix least-squares treatment against $|F|^2$ in anisotropic approximation with SHELX 2014/7 [96] in the ShelXle program [97]. H-atoms were refined in the geometrically calculated positions. The crystallographic data have been deposed in the Cambridge Crystallographic Data Centre under the deposition codes CCDC 2182022.

**Computational details.** The full geometry optimization procedure for all model structures and DFT calculations for model supramolecular associates based on the obtained experimental X-ray geometry of **2** have been carried out using the dispersion-corrected hybrid functional ωB97XD [98] with the help of Gaussian-09 [99] program package. The Douglas–Kroll–Hess 2nd order scalar relativistic calculations requested for the relativistic core Hamiltonian were carried out using the DZP-DKH basis sets [100–103] for all atoms. No symmetry restrictions were applied during the geometry optimization procedure. The Hessian matrices were calculated analytically for all optimized model structures to confirm the localization of true minima on the potential energy surface (no imaginary frequencies were found in all cases). The topological analysis of the electron density distribution has been performed using the Multiwfn program (version 3.7) [104]. The Cartesian atomic coordinates for all optimized equilibrium model structures and model supramolecular associates based on the obtained experimental X-ray geometry of **2** are presented in Tables S2 and S3 (Supplementary Materials).

**Synthesis of [CuBr₂(4-Me-Ph-bian)] (1).** A mixture of $CuBr_2$ (224 μmol, 50 mg) and 4-Me-Ph-bian (224 μmol, 80.7 mg) in 8 mL of acetonitrile was refluxed for 3 h. The resulting dark brown solution with the precipitate was evaporated to dryness and the residue was washed with hexane. Yield: 102 mg (78%). Anal. Calc. for $C_{26}H_{20}Br_2CuN_2$: C 53.4, H 3.5, N 4.8%; Found C 52.8, H 3.7, N 4.7%. IR (KBr) ν/cm$^{-1}$: 3453 (br. vw), 3046 (w), 2976 (w), 2947 (w), 2919 (w), 2863 (w), 1661 (w), 1624 (w), 1587 (s), 1505 (s), 1486 (m), 1435 (m), 1422 (vs), 1381 (w), 1363 (w), 1314 (vs), 1288 (w), 1251 (m), 1227 (w), 1217 (w), 1186 (s), 1151 (w), 1135 (w), 1109 (vs), 1044 (m), 1022 (w), 981 (w), 933 (w), 829 (vs), 774 (vs), 742 (w), 711 (w), 701 (w), 652 (w), 636 (m), 616 (w), 589 (w), 553 (m), 527 (w), 488 (w), 450 (w), 419 (w).

**Synthesis of [CuBr₂(2-Me-Ph-bian)] (2).** A mixture of $CuBr_2$ (224 μmol, 50 mg) and 2-Me-Ph-bian (224 μmol, 80.7 mg) in 8 mL of acetonitrile was refluxed for 3 h. The resulting dark brown solution with the precipitate was evaporated to dryness and the residue was washed with hexane. Dark brown crystals were obtained by layering hexane on a solution of complex **2** in dichloromethane. Yield: 115 mg (88%). Anal. Calc. for $C_{26}H_{20}Br_2CuN_2$: C 53.4, H 3.5, N 4.8%; Found C 53.7, H 3.5, N 4.8%. IR (KBr) ν/cm$^{-1}$: 3425 (br. vw), 3056 (m), 3018 (w), 2976 (vw), 2910 (w), 2857 (w), 1634 (vs), 1582 (vs), 1482 (s), 1460 (w), 1446 (w), 1434 (w), 1418 (m), 1378 (w), 1358 (w), 1293 (m), 1251 (w), 1226 (w), 1188 (w), 1153 (w), 1132 (w), 1121 (w), 1093 (w), 1049 (m), 954 (w), 940 (w), 859 (w), 835 (s), 777 (vs), 756 (vs), 717 (s), 627 (w), 615(w), 543 (m), 503(w), 445(w), 413 (w).

## 4. Conclusions

Two novel Cu(II) complexes with redox-active BIAN ligands [CuBr₂(R-bian)] (R = 4-Me-Ph (**1**), 2-Me-Ph (**2**)) were obtained in good yield by a ligand exchange reaction from copper(II) bromide. The complexes are structural isomers and differ only in the position of the methyl group in BIAN. For complex **2**, the crystal structure was determined by X-ray diffraction analysis. The coordination environment of the Cu(II) center is distorted square-planar. On the contrary, the $CuBr_2N_2$ geometry in structure **1** is closer to a distorted tetrahedron, according to the DFT calculations. To the best of our knowledge, complexes

**1** and **2** are the first examples of copper complexes with Me-Ph-bian ligands. The redox properties of complexes **1** and **2** in CH$_3$CN were studied using cyclic voltammetry. The coordination of the redox-active Me-Ph-bian ligand to the Cu(II) center leads to both the reversibility of the Cu(II)/Cu(I) process and a rather high oxidizing ability of Cu(II), providing the E$_{1/2}$ potentials of 0.81 and 0.66 V (vs. SHE) for **1** and **2**, respectively. Accordingly, a partial reduction of Cu(II) to Cu(I) in solution was detected using UV-Vis spectroscopy. The differences in redox potentials can be related to the different structure of the CuBr2N2 coordination site. Both complexes **1** and **2** were tested as homogeneous catalysts for IPB oxidation. During the catalytic reaction, the ratio of Cu(II) and Cu(I) species differed from the initial one in the CH$_3$CN solution. That was consistent with the radical chain oxidation initiated with the Cu(II)/Cu(I) couple. Simultaneously, complexes **1** and **2** were able to decompose IPBHP with the formation of RO• radicals without changing the oxidation state of Cu(II). The decomposition of IPBHP catalyzed by **1** or **2** resulted in the formation of products of homolytic cleavage of the O-O bond, which indicated the possibility of a Cu(II)-IPBperoxide adduct forming as an intermediate.

**Supplementary Materials:** The following supporting information can be downloaded at: https://www.mdpi.com/article/10.3390/catal13050849/s1, Table S1: Crystal data and structure refinement for **2**; Table S2: Cartesian atomic coordinates for all optimized equilibrium model structures; Table S3: Cartesian atomic coordinates for model supramolecular associate based on the obtained experimental X-ray geometry of **2**; Table S4: Conversion of IPBHP and products obtained in CH$_3$CN solutions of **1** and **2**. Conditions: 0.001 mmol of **1** or **2**, 4 mmol of IPBHP in 2.5 mL CH$_3$CN, 30 °C, 130 min, Ar; Figure S1: Formation of π-π interactions between the dimers in the crystal structure of **2**; Figure S2: Crystal packing of **2**; Figure S3: UV-Vis diffuse reflectance spectra of complexes **1** and **2** in solid state; Figure S4: UV-Vis absorption spectra of CH$_3$CN solutions of complexes **1** and **2** ($5 \times 10^{-4}$ M); Figure S5: Cyclic voltammograms of **1** (A) and **2** (B) in CH$_3$CN in the −2.3–2.0 V region (black line) and the 0–1.0 V region (blue line) at a potential scan rate of 100 mV/s; Figure S6: Kinetic curves of oxygen uptake by CuBr$_2$ (A) and CuCl$_2$ (B) solutions. Conditions: 1 mL of IPB, 1.5 mL of CH$_3$CN, $5 \cdot 10^{-4}$ M of catalyst; Figure S7: UV-Vis spectra of **1** (A) and **2** (B) ($3 \times 10^{-4}$ M) in CH$_3$CN solution (1), 10 min after addition of IPBHP (2), 40 min after addition of IPBHP (3).

**Author Contributions:** I.S.F.: Investigation. Synthesis and description of copper complex compounds; O.S.K.: Investigation, Writing—original draft preparation. Investigation on catalysis of copper complexes. Description of catalytic data; N.I.K.: Investigation, Writing—original draft. Carrying out catalytic reactions catalyzed by copper complexes. Generalization and description of catalytic data; T.V.L.: Investigation. Conducting a spectroscopic study of copper complexes; M.I.G.: Investigation. Synthesis and purification of ligands; M.A.: Investigation. Crystallization and purification of copper complexes; P.A.A.: Investigation. Interpretation of X-ray diffraction data; A.S.N.: Investigation. DFT calculations; A.L.G.: Conceptualization, Writing—review & editing. Writing and editing the first draft of the article. Communication with the editor of the journal. All authors have read and agreed to the published version of the manuscript.

**Funding:** Financial support from the Russian Science Foundation (grant No. 22-23-20123) and the government of the Novosibirsk region (contract r-39) is acknowledged.

**Data Availability Statement:** The data presented in this study are available on request from the corresponding author.

**Acknowledgments:** The DFT calculations and topological analysis of the electron density distribution were supported by the RUDN University Strategic Academic Leadership Program. The authors thank the Ministry of Science and Higher Education of the Russian Federation, the Centre of Collective Usage of NIIC SB RAS for the XRD data collection as well as the shared research center "National center of investigation of catalysts" of BIC SB RAS for the UV-Vis studies.

**Conflicts of Interest:** The authors declare no conflict of interest.

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
