# Peer review of "Novel Copper(II) Complexes with BIAN Ligands: Synthesis, Structure and Catalytic Properties of the Oxidation of Isopropylbenzene"

_catalysts, doi:10.3390/catal13050849_

Round 1
Reviewer 1 Report
A synthesis of two new copper complexes and their reactivity in an oxidation reaction is reported by Gushchin and coworkers. BIAN ligands are used to make the new copper complexes, and their structures are elucidated through several spectroscopic techniques. The paper is well-written and is a good study of these two complexes. The authors outline several points in the introduction which describe the novelty and importance of the work. I agree with these points which include that oxidation reactions with BIAN ligands are limited to Vanadium. This seems to be the first study on radical chain oxidations with Copper(II) BIAN complexes, and the first example of a C-H oxidation with this ligand type. While the reactivity is not very impressive (e.g., low selectivity and scope limited to one substrate), this is fine since it is not the focal point of the paper. The low selectivity is not surprising given that the catalyst is not directly involved in selectivity determining steps (i.e., C-H cleavage or functionalization). Below are a couple of points the authors should comment on in the manuscript.
1. Given the similarities in complexes 1 and 2, it is surprising that one is highly soluble in DCM and ACN, while the other is not. What is the solubility? The authors should provide data to support this statement.
2. Chemically, is there any reason why complex 1 would participate in MLCT to a greater extent than 2? Also, why does complex 1 have a higher oxidizing ability?
Aside from these, the manuscript is publishable as is and the manuscript does not need to be returned to me for additional review.
Author Response
1. Given the similarities in complexes 1 and 2, it is surprising that one is highly soluble in DCM and ACN, while the other is not. What is the solubility? The authors should provide data to support this statement.
Despite the fact that the ligands differ only in the position of the methyl group, we believe that this significantly affects the packing energy in the crystal, which leads to different solubility of complexes 1 and 2. The solubility of the complexes in acetonitrile is approx. 1 mg/ml for 2 and 0.3 mg/ml for 1.
2. Chemically, is there any reason why complex 1 would participate in MLCT to a greater extent than 2? Also, why does complex 1 have a higher oxidizing ability?
The charge transfer band (MLCT), that appears in the UV-Vis spectra of 1 and 2 in acetonitrile, is associated with the formation of copper(I) species due to partial Cu(II)/Cu(I) reduction. The CV data showed that the Cu(II)/Cu(I) redox potential is higher for complex 1. In other words, complex 1 is a stronger oxidant than complex 2. Thus, complex 1 is more readily reduced in solution and the MLCT band is more pronounced in this case.
Reviewer 2 Report
This research article by Prof. Artem L. Gushchin and coworkers reported the first time preparation of two novel Cu(II)-BIAN complexes from CuBr2 with good yields. The author conducted extensive studies to analyze the structure and activity of the above two complexes, such as x-ray diffraction analysis, DFT calculations, UV-Vis spectra and Cyclic voltammetry. Interestingly, although complex 1 and 2 possess similar structure, the structure of 2 is a distorted square-planar the structure of 1 is distorted tetrahedron. Importantly, the catalytic studies indicated that both catalysts are good homogeneous catalysts and show excellent performance in IPB oxidation via radical pathway. The manuscript is well organized and written. The manuscript deserves to be published in Catalysts after minor revision.
1) The author synthesized two Cu(II)-BIAN complexes, after synthesis the author conducted Anal. Calc. and IR (KBr) analysis. Although the structure of complex 2 has been confirmed by x-ray diffraction analysis and complex 1 been analyzed by DFT calculations, I suggest the author conduct NMR analysis and provide NMR information for the above two compounds. The NMR information usually is one of the most important structure information for new compounds.
2) Page 10, first paragraph: “chain mechanism 1-5,” suggest change to “chain mechanism (eq. 1-5),”. Same, in page 9, change the number “1-5” to “eq.1 eq.2 ...”. Same change suggested on page 11.
3) The conclusion has 3 paragraphs, suggesting to concise the contents to one paragraph.
4) In the reference, the author should be very careful about the format. For example: 1) the journal name should use the abbreviation, such as reference 5 “Chemistry - A European Journal” should be “Chem. Eur. J.” same correction to other references; 2) the abbreviation should be end by “.”, such as reference 8 “Inorg Chem” should be “Inorg. Chem.”; 3) add space between abbreviations of the author's name, such as reference 11, “Fedushkin, I.L.;” should be “Fedushkin, I. L.;” same correction to all references.
Author Response
1. The author synthesized two Cu(II)-BIAN complexes, after synthesis the author conducted Anal. Calc. and IR (KBr) analysis. Although the structure of complex 2 has been confirmed by x-ray diffraction analysis and complex 1 been analyzed by DFT calculations, I suggest the author conduct NMR analysis and provide NMR information for the above two compounds. The NMR information usually is one of the most important structure information for new compounds.
NMR spectra for complexes 1 and 2 were not recorded, since the Cu(II) complexes are paramagnetic (d9 ion). The presence of unpaired electrons leads to a shift and broadening of signals in the NMR spectra, especially in the region of aromatic protons.
2. Page 10, first paragraph: “chain mechanism 1-5,” suggest change to “chain mechanism (eq. 1-5),”. Same, in page 9, change the number “1-5” to “eq.1 eq.2 ...”. Same change suggested on page 11.
Corrected
3. The conclusion has 3 paragraphs, suggesting to concise the contents to one paragraph.
Сorrected
4. In the reference, the author should be very careful about the format. For example: 1) the journal name should use the abbreviation, such as reference 5 “Chemistry - A European Journal” should be “Chem. Eur. J.” same correction to other references; 2) the abbreviation should be end by “.”, such as reference 8 “Inorg Chem” should be “Inorg. Chem.”; 3) add space between abbreviations of the author's name, such as reference 11, “Fedushkin, I.L.;” should be “Fedushkin, I. L.;” same correction to all references.
Corrected